# CLiF-VQA: Enhancing Video Quality Assessment by Incorporating High-Level Semantic Information related to Human Feelings

### Yachun Mi
Harbin Institute of Technology
Harbin, China
miyachun@stu.hit.edu.cn

### Yan Shu
Harbin Institute of Technology
Harbin, China
shuyan_hit@163.com

### Yu Li
Harbin Institute of Technology
Harbin, China
lyhit2001@gmail.com

### Chen Hui
Harbin Institute of Technology
Harbin, China
20b903013@stu.hit.edu.cn

### Puchao Zhou
Harbin Institute of Technology
Harbin, China
a964989182@gmail.com

### Shaohui Liu*
Harbin Institute of Technology
Harbin, China
shliu@hit.edu.cn

## ABSTRACT

Video Quality Assessment (VQA) aims to simulate the process of perceiving video quality by the Human Visual System (HVS). Although subjective studies have shown that the judgments of HVS are strongly influenced by human feelings, it remains unclear how video content relates to human feelings. The recent rapid development of Vision-Language pre-trained models (VLM) has established a solid link between language and vision. And human feelings can be accurately described by language, which means that VLM can extract information related to human feelings from visual content with linguistic prompts. In this paper, we propose CLiF-VQA, which innovatively utilizes the visual linguistic capabilities of VLM to introduce human feelings features based on traditional spatio-temporal features to more accurately simulate the perceptual process of HVS. In order to efficiently extract features related to human feelings from videos, we pioneer the exploration of the consistency between Contrastive Language-Image Pre-training (CLIP) and human feelings in video perception. In addition, we design effective prompts, i.e., a variety of objective and subjective descriptions closely related to human feelings, as prompts. Extensive experiments show that the proposed CLiF-VQA exhibits excellent performance on several VQA datasets. The results show that introducing human feelings features on top of spatio-temporal features is an effective way to obtain better performance.

## CCS CONCEPTS

• **Computing methodologies** → **Computer vision tasks**.

## KEYWORDS

Video Quality Assessment, Human Feelings, Vision-Language, Semantic Information, Deep Learning

---

*Corresponding author.

**ACM Reference Format:**
Yachun Mi, Yan Shu, Yu Li, Chen Hui, Puchao Zhou, and Shaohui Liu. 2024. CLiF-VQA: Enhancing Video Quality Assessment by Incorporating High-Level Semantic Information related to Human Feelings. In *Proceedings of the 32nd ACM International Conference on Multimedia (MM '24), October 28-November 1, 2024, Melbourne, VIC, Australia.* ACM, New York, NY, USA, 10 pages. https://doi.org/10.1145/3664647.3680930

## 1 INTRODUCTION

With the rapid advancement of technology, the threshold of video production has been significantly lowered, enabling an increasing number of users to create and upload videos to various online platforms. However, User-Generated Content (UGC) videos often have annoying distortion because of the absence of professional filming equipment and skills. Moreover, compression techniques [20, 22, 68, 74] and copyright protection processing [21] can also damage the quality of UGC videos. Therefore, Video Quality Assessment (VQA) of in-the-wild videos is increasingly important for major video platforms to filter out and enhance low-quality videos.

The lack of raw information and the diversity of distortion types in in-the-wild videos present a significant challenge for VQA research. Fortunately, there are many subjective experiments that provide high-quality datasets [16, 32, 38, 43, 48, 56, 70], which are labeled according to human mean opinion scores (MOS). With the benefit of these datasets, the current VQA methods can perform supervised training on them to fit the MOS as best as possible. Traditional VQA methods [1, 5, 8, 25, 33, 41, 46, 53] are successful in predicting the quality of perceptual videos, which model spatial and temporal distortions using handcrafted features. However, the hand-crafted features have a low correlation with human perception, so its outcomes are not always reliable. In recent years, with the advancement of deep learning techniques, VQA methods [28, 29, 49, 59, 70] based on deep neural networks (DNNs) can extract more complex and abstract features related to video quality and achieve superior performance than traditional methods. However, most deep learning approaches focus on the effect of spatial and temporal video distortion on video quality, without adequately considering the relationship between video quality factors and human feelings. Research has shown that human judgment is always influenced by how the brain feelings [39, 72]. As shown in Fig. 1, two videos with the same objective quality but different content have different subjective quality scores. Considering that all the

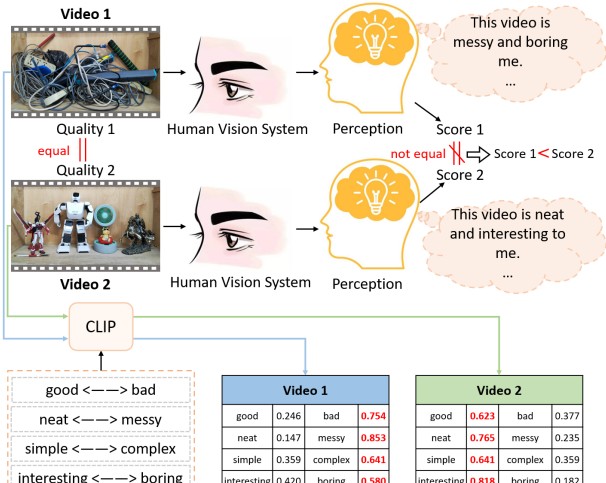

**Figure 1: Validating the impact of human feelings on HVS for VQA and the relevance of CLIP with human feelings in video perception. Two videos with the same quality captured in the same scene using the same equipment. CLIP show high consistency with human perception in video perception. We selected 10 subjects to perform VQA on the two videos and took their mean value as the video quality score.**

current datasets used for VQA are labeled based on HVS, incorporating human feelings into VQA enables the model to achieve better consistency with HVS. Although some recent studies [9, 29, 57, 60] have demonstrated that video content can indeed influence the human judgment of the video quality, these investigations predominantly extract high-level abstract features that are not directly related to human intuitive perception to assess the impact of video content on HVS. Therefore, we believe that these features do not effectively capture the true human feelings of video quality, thus limiting the effectiveness of these methods in practical applications.

Extracting features that capture human feelings from videos presents a significant challenge. This is largely due to the complexity and subtlety of feelings conveyed through videos, which are not easily encapsulated by straightforward data features. It is well known that human feelings can be accurately described through language. Therefore, if we can deduce the feelings a video induces through its associated linguistic expressions, we may open new pathways to tackle the intricacies of such nuanced feature extraction, thereby potentially resolving the challenges previously delineated. Fortunately, the recent advancements in Vision-Language pre-trained models (VLM) have significantly propelled the field forward. These models have not only established a crucial link between linguistic and visual information but have also equipped us with the ability to interpret the feelings elicited by video content through an analysis of language expressions. Specifically, Contrastive Language-Image Pre-training (CLIP) [44], endowed with a rich vision language prior, has demonstrated robust zero-shot predictive capabilities across multiple tasks. Furthermore, CLIP has the ability to perceive image and video quality to some extent [37, 55]. Nonetheless, a pending challenge in video perception is whether CLIP has good agreement with human feelings. If so, this implies that by conducting an in-depth analysis of the linguistic descriptions related to videos, we

can capture the feelings intended to be conveyed by the videos with greater precision, thereby making unprecedented strides in the understanding and interpretation of these feelings.

In this paper, to address the above difficulty, we verify through extensive experiments that CLIP has a high degree of consistency with human feelings in video perception. Further, we propose a novel model (denoted as CLiF-VQA) to enhance video quality assessment (VQA) by incorporating high-level semantic information related to human feelings. Our model innovatively utilizes the visual language capabilities of CLIP to extract features from visual content that are relevant to human feelings. In addition, it captures low-level-aware features by using the Video Swin Transformer model [36] to reflect spatial and temporal distortions in video frames, providing a comprehensive framework for assessing video quality that is highly consistent with human perception. Specifically, we use a set of objective (e.g., bright, blurry, noisy, colorful, etc.) and subjective (e.g., pleasant, boring, fearful, etc.) descriptions that are closely related to human feelings as prompts. The cosine similarity between the visual content and the text prompts is then computed thereby predicting the score corresponding to each prompt. Further, we design a semantic feature extractor (SFE), which extracts high-level semantic feature maps corresponding to descriptions from multiple regions of the video frame. Finally, we fuse the low-level-aware and high-level semantic features to obtain the video quality score.

Our contributions can be summarized as follows:

- **We validate for the first time that CLIP is highly consistent with human feelings in video perception.**
- **We propose CLiF-VQA, which for the first time incorporates features related to human feelings in VQA.** Extensive experiments demonstrate that CLiF-VQA achieves the best performance on multiple VQA datasets.
- **We design some efficient objective and subjective descriptions that are related to human feelings**. These prompts enable us to extract from the video rich features related to subjective and objective human feelings.
- **We design a zero-shot advanced semantic feature extractor (SFE) based on CLIP.** It extracts semantic features by sliding over multiple regions of a video frame and splices the same semantic features according to their relative positions to obtain semantic feature maps of the video frame.

## 2 RELATED WORK

### 2.1 Classical VQA Methods

Classical VQA methods employ handcrafted features to capture specific types of distortions in the video for quality prediction. Early VQA often apply Image Quality Assessment (IQA) algorithms [12, 27, 40, 42, 67, 69] to obtain frame-level features, and then combine with temporal dimension information to obtain video quality scores. For example, V-CORNIA [66] extends the IQA algorithm CORNIA [69] to VQA to obtain frame-level quality scores, and combines these scores through temporal pooling. However, this method does not fully consider the connection between the spatio-temporal information of the video and how they affect the video quality [2, 23, 45, 46]. Natural video statistics (NVS) can take into account both spatio and temporal information, thus it is applied to address the previous problem. V-BLIINDS [46] extracts spatio-temporal

statistical features of frame-differences in the video DCT domain
and predicts crude frame quality scores using NIQE [42]. TLVQM
[25] considers two levels of features, first computing low complexity
features for each frame to extract frame-level statistical features
related to motion, and then computing high complexity features
related to spatial distortion for representative frames. VIDEVAL
[53] applies various handcrafted features to detect and measure the
distortions and reduces the computational complexity by reducing
the feature dimensions.

## 2.2 Deep Learning-based VQA Methods

Recently, deep learning-based VQA Methods [3, 28, 29, 34, 49, 58–
62, 65, 70, 73, 75, 76] have gradually achieved better performance
than classical methods. Rather than relying on handcrafted features,
deep VQA methods employ convolutional neural networks (CNN)
[10, 11, 13, 14, 47, 50–52] or Transformer models [7, 35, 36] to ex-
tract complex and abstract features that are relevant to video quality
aspects. For example, VSFA [29] extracts spatial features of video
frames using ResNet-50 [13] pre-trained on ImageNet [6], and then
models the temporal features using GRU [4]. Similar to the archi-
tecture of VSFA, while GST-VQA [3] applies VGG-16 [47] to extract
spatial features of videos. To better capture the spatio-temporal
information of the video, some works [34, 49, 57, 70, 71, 73] adopt
3D-CNN. For example, V-MEON [34] adopts a multi-task frame-
work which utilizes 3D-CNN to extract spatio-temporal features to
predict the quality of the video. Other studies [49, 57, 70, 73] com-
bine both 2D-CNN and 3D-CNN to capture the spatial and temporal
features of video, and then integrate the two features for quality
prediction. Recently, VQA methods [58–60] using the transformer
structure have achieved better results relative to CNN. DisCoVQA
[60] uses Video Swin Transformer [36] to extract multi-level spatio-
temporal features and improves the learning efficiency of the model
by temporal sampling of the features. Similarly, FAST-VQA [58] and
FasterVQA [59] obtain fragments by spatial-temporal grid mini-
cube sampling (St-GMS) and then feed the fragments into a modified
Video Swin Transformer [36]. Although deep learning-based VQA
methods can extract complex high-level semantic features, these
features are not directly related to the human point of view. Two
recent works [63, 64] attempt to address this issue. Specifically,
MaxVQA [64] captures a variety of quality factors that can be ob-
served by humans through a modified vision-language foundation
model CLIP and can jointly evaluate multiple specific quality factors
and overall perceptual quality scores. Several studies [30, 57, 60]
have noted that aesthetic factors [15, 17–19, 31] of visual content af-
fect video quality assessment. Inspired by this, Dover [63] assesses
video quality from both aesthetic and technical perspectives, so it
relatively well models the human process of perceiving quality.

## 3 CLIP FOR VIDEO PERCEPTION

CLIP, as shown in Fig. 2, demonstrates excellent zero-shot predic-
tion ability in vision-language tasks. Not only that, it also shows
some perceptive ability in IQA and VQA [37, 55]. However, it is not
verified whether it still has good perceptual ability on linguistic
prompts related to human feelings. The study in this section repre-
sents a pioneering effort to ascertain the degree to which the CLIP's
video perception aligns with human feelings, thereby ensuring the

extraction of human affective features from videos with the highest
possible fidelity. Specifically, in order to fully extract video features
while avoiding quality loss due to resizing and cropping, we extract
semantic features from multiple regions on all video frames by
means of sliding window (Details in Sec. 4.1). Then we compute the
mean of all the feature values corresponding to a specific prompt
as the feature of the video for that prompt.

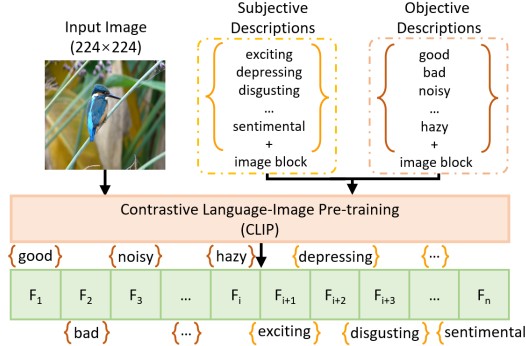

**Figure 2: The process of extracting semantic features using
CLIP. Each feature $F_i$ corresponds to a description.**

**Prompt Design.** We apply multiple objective descriptions related
to quality factors (e.g., bright, contrast, etc.) and multiple subjective
descriptions related to human feelings (e.g., interesting, exciting,
etc.) as prompts. For details see Sec. 4.1. Relative to the antonym
prompts strategy [55], our design can extract richer features related
to human feelings from videos. Here, we refer to HVS's percep-
tion of video quality factors and content as human objective and
subjective feelings respectively. Further, we explore the correlation
between CLIP and human objective feelings and subjective feelings.

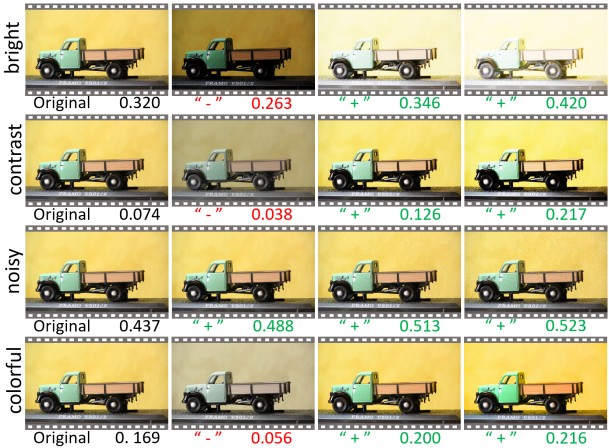

**Figure 3: CLIP for perception of four objective descriptions
(bright, contrast, noisy, colorful). "-" represents attenuation
and "+" represents enhancement.**

**Perception of Objective Feelings.** We explore the performance
of CLIP on four objective descriptions (bright, contrast, noisy, color-
ful) related to video quality factors, as shown in Fig. 3. Specifically,
we first process the video corresponding to a certain description,
and then extract the semantic features that correspond to the de-
scription. It can be seen that CLIP is able to accurately perceive

changes in video quality factors. This shows that CLIP has a good consistency with human objective feelings in the perception of video quality factors.

**Perception of Subjective Feelings.** Furthermore, we explore the relationship between CLIP and human subjective feelings in video content perception. In particular, we conduct experiments on four subjective descriptions (interesting, exciting, depressing, fearful) that reflect the subjective feelings that video content brings to humans. As shown in Fig. 4. The results show that CLIP is highly consistent with human judgments in perceiving video content.

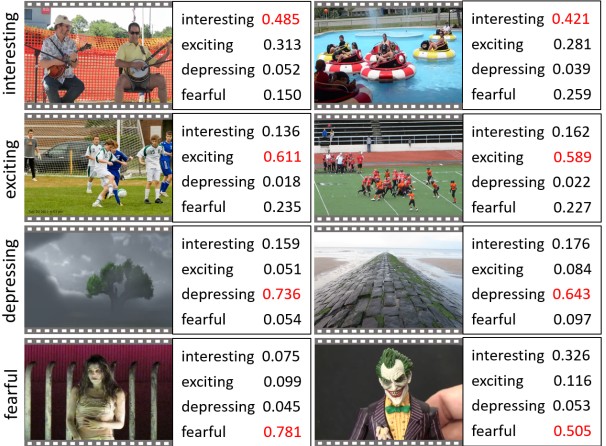

**Figure 4: CLIP for perception of four subjective descriptions (interesting, exciting, depressing, fearful).**

**Performance of CLIP in VQA.** The experiments above demonstrate that CLIP has highly consistent results with humans in perceiving both the quality and content of the video separately. However, it remains to be verified whether CLIP is still effective when both objective and subjective descriptions are used as prompts. Therefore, we conduct further experiments to explore CLIP's performance in video quality perception when using both objective and subjective descriptions that can reflect human feelings.

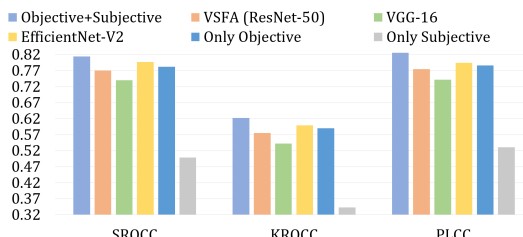

**Figure 5: Comparison results of CLIP with some classical methods in VQA.**

Specifically, we use multiple objective and subjective descriptions as prompts, as shown in Fig. 2. We adopt the architecture of the classical VQA model VSFA [29] to input the feature vectors of the frames into the GRU for regression and time pooling operation to get the quality scores. We conduct our experiments on KoNViD-1k dataset [16], and in addition to processing the comparison with the VSFA model, we also compare with two feature extraction methods (VGG-16 [47] and EfficientNet-V2 [51]) widely used in VQA. And

we evaluate performance on SROCC, KROCC and PLCC metrics, as shown in Fig. 5. The results demonstrate that using both objective and subjective descriptions can achieve better results, compared to using a single description. Furthermore, it can be observed that relying on only features related to human feelings surpasses CNN extracted features in VQA. The results also confirm the validity of the prompts we designed.

## 4 THE PROPOSED APPROACH

In this section, we introduce the proposed CLiF-VQA, which consists of the semantic feature extraction module 4.1 and the spatial feature extraction module 4.2, as shown in Fig. 6. First, we employ the semantic feature extraction module to extract high-level semantic features that are related to human feelings. Then low-level-aware features are extracted using spatio-temporal feature extraction module. Finally we fuse these two features through a regression module thus obtaining the video quality score.

### 4.1 Semantic Feature Extraction

In order to effectively extract features that can reflect human feelings, we first design some objective descriptions and subjective descriptions related to human feelings as prompts of CLIP, as shown in Fig. 2. The prompts $P$ designed in this paper contains two types of descriptions: objective $p^{ob}$ and subjective $p^{sub}$:

$$P = [p_1^{ob}, p_2^{ob}, ..., p_{n_1}^{ob}, p_1^{sub}, p_2^{sub}, ..., p_{n_2}^{sub}] \quad (1)$$

The adjectives and nouns that delineate feelings within objective descriptions are denoted as $A^{ob}$ and $N^{ob}$ respectively. Objective descriptions are categorized into the following two forms depending on whether they use adjectives or nouns:

$$p_i^{ob} = \begin{cases} A_{i'}^{ob} + \text{"image block"} \\ \text{"image block with"} + N_{i'}^{ob} \end{cases}, 1 \le i' \le n_1 \quad (2)$$

Subjective descriptions exclusively employ adjectives $A^{sub}$ to convey feelings:

$$p_{j'}^{sub} = A_{j'}^{sub} + \text{"image block"}, 1 \le j' \le n_2 \quad (3)$$

In addition, due to the limitation of the visual encoder of CLIP on the input size, we can only extract the semantic information of a small region in the video frame. In order to be able to obtain as much semantic information as possible contained in the video frames, we extracted features from multiple regions of the video frames by sampling them multiple times at different locations, as shown in Fig. 6(b). This avoids the loss of video quality caused by resizing and excessive cropping.

Specifically, assuming the video has $T$ frames, we perform a sampling operation on the video frames $I_t$ ($t = 1, 2, ..., T$) to obtain $m \times n$ image blocks:

$$\left\{ b_t^{i,j} | 1 \le i \le m, 1 \le j \le n \right\} = Sampling(I_t) \quad (4)$$

where $b_t^{i,j}$ represents the block obtained by sampling in the i-th row and j-th column.

Given any visual input $b_t^{i,j}$ and text prompt $P$, the vision $E_v$ and text $E_t$ encoders of CLIP encode them to achieve a consistent

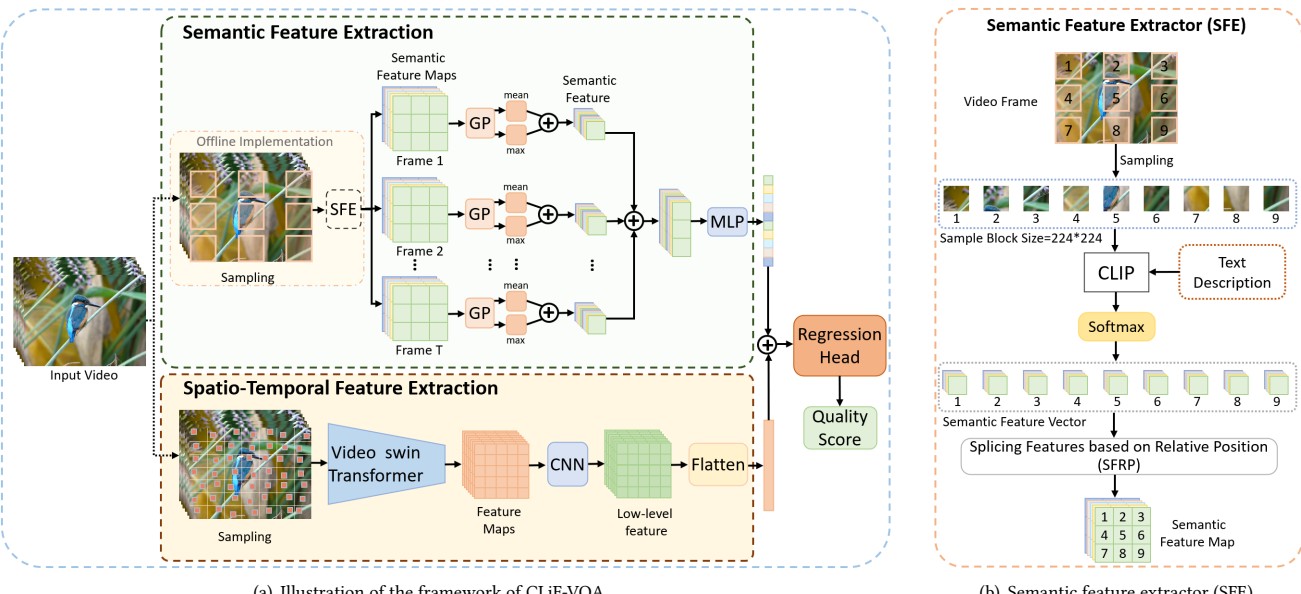

(a) Illustration of the framework of CLiF-VQA.

(b) Semantic feature extractor (SFE).

**Figure 6: The framework of CLiF-VQA, which extracts semantic features related to human feelings through the semantic feature extraction module as well as low-level-aware features through the spatio-temporal feature extraction module, and then obtains the quality scores by aggregating the two features through the aggregation header.**

representation within a unified feature space:

$$f_{v,t}^{i,j} = E_v(b_t^{i,j}); f_t = E_t(P) \tag{5}$$

Then, calculate the cosine similarity between the visual content and prompts to predict the score for each dimension:

$$Sim(b_t^{i,j}, P) = s_{v,k}^{i,j}|_{k=1}^{n_1+n_2} = \frac{f_{v,t}^{i,j} \cdot f_t}{\left\|f_{v,t}^{i,j}\right\| \|f_t\|} \tag{6}$$

A normalization procedure is applied to the acquired cosine similarity score to standardize its range:

$$v_t^{i,j} = \frac{exp(s_{t,l}^{i,j})}{\sum_{k=1}^{n_1+n_2} exp(s_{t,k}^{i,j})}|_{l=1}^{n_1+n_2} \tag{7}$$

where the feature values of $v_t^{i,j}$ are the same number and one-to-one correspondence with the number of descriptions.

Then the splicing operation (SFRP) is performed on all the features $v_t^{i,j}$ based on the relative position to obtain the semantic feature map $M_t$ of frame $I_t$:

$$M_t = SFRP(\{v_t^{i,j}|1 \le i \le m, 1 \le j \le n\}) \tag{8}$$

$M_t$ contains $r$ feature maps, each corresponding to a description. Further, we perform the global average pooling operation ($GP_{avg}$) and the global max pooling operation ($GP_{max}$) on $M_t$ to obtain the universal features and distinctive features as shown Fig. 6(a). The outputs of ($GP_{avg}$) and ($GP_{max}$) are two r-dimensional feature vectors $f_t^{avg}$ and $f_t^{max}$, respectively.

$$f_t^{avg} = GP_{avg}(M_t), f_t^{max} = GP_{max}(M_t) \tag{9}$$

$f_t^{avg}$ and $f_t^{max}$ are then concatenated as the semantic feature vectors $f_t$ of the video frame $I_t$:

$$f_t = f_t^{avg} \oplus f_t^{max} \tag{10}$$

where $\oplus$ is the concatenation operator and $f_t$ is a feature vector of length $2 \times (n_1 + n_2)$.

Next, we perform a concatenation operation on the semantic features $\{f_t\}_{t=1}^T$ of all the video frames thereby obtaining the semantic feature maps $M_s$ of the video:

$$M_s = f_1 \oplus f_2 \oplus f_3 \oplus ... \oplus f_T \tag{11}$$

where $\oplus$ here is not exactly the same as the concatenation of $\oplus$ in Eq. 10. Here, the feature vectors $\{f_t\}_{t=1}^T$ are concatenated along the channel dimension, so that the dimension of the obtained $M_s$ is $[2 \times (n_1 + n_2), T]$. Each feature map corresponds to a description, and the feature maps here are divided into two types, namely the feature map with global average pooling operation ($GP_{avg}$) and the feature map with global max pooling operation ($GP_{max}$).

After extracting the semantic feature maps of the video, we use a multi-layer perceptron (MLP) to obtain the feature vectors $F_s$ corresponding to the descriptions. The MLP is composed of two fully connected layers and the activation function is GELU:

$$F_s = FC_2(GELU(FC_1(M_s))) \tag{12}$$

where $F_s$ is a $2 \times (n_1 + n_2)$ dimensional feature vector.

### 4.2 Spatio-Temporal Feature Extraction

In VQA, the spatio-temporal features of the video play a very important role in estimating the overall video quality. Since low-level information is easily affected by distortion, extracting the low-level-aware features of the video can effectively capture the spatio-temporal distortion of the video.

In our approach, to diminish the computational complexity, the video is sampled employing the grid mini-patch sampling approach [58]. First, the video frame $I_t (t = 1, 2, ..., T)$ is segmented into $N \times N$

grids of equal size:

$$g_t^{i,j} = I_t\left[\frac{i \times H}{N} : \frac{(i+1) \times H}{N}, \frac{j \times W}{N} : \frac{(j+1) \times W}{N}\right] \quad (13)$$

where $g_t^{i,j}$ denotes the grid of the i-th row and j-th column, and W and H are the height and width of the video.

A random patch sampling is then performed on each $g_t^{i,j}$ thus obtaining a mini-patch $MP_t^{i,j}$:

$$MP_t^{i,j} = Sampling_t^{i,j}(g_t^{i,j}) \quad (14)$$

where $Sampling_t^{i,j}$ represents the random sampling operation on the grid of the i-th row and j-th column of the t-th frame. The sampling operation samples the same position on different video frames to ensure temporal continuity.

Then splice $MP_t^{i,j}(1 \leq i, j \leq N)$ according to their original positions thereby obtaining the sampled map $S_t$ of the video frame $I_t$. The same operation is performed on all frames of the video to obtain the sampled fragments $V_f = [\{S_t|_{t=1}^T\}]$.

The video fragments $V_f$ are then fed into a modified Video Swin Transformer Tiny [36] and non-linear layers to obtain local quality maps $M_{final}$.

$$M_f = Swin\ Transformer(V_f) \quad (15)$$

Finally we flatten $M_{final}$ to obtain the spatio-temporal feature vector $F_f$ of the video.

$$F_f = Flatten(M_{final}) \quad (16)$$

### 4.3 Quality Regression

After extracting the semantic and spatio-temporal features of the video through the semantic feature extraction module and spatio-temporal feature extraction module, we need to map these features to the quality scores via a regression model. First, We concatenate the semantic features $F_s$ and spatial features $F_f$ to get the overall features $F_v$ of the video:

$$F_v = F_s \oplus F_f \quad (17)$$

Then we design a regression head with two fully connected layers to predict the quality score of the video:

$$Score = FC_4(GELU(FC_3(F_v))) \quad (18)$$

### 4.4 Loss Function

The loss function used to optimize the proposed models consists of two parts: the monotonicity-induced loss and linearity-induced loss. Given m predicted quality scores $\hat{Q} = \{\hat{q_1}, \hat{q_2}, ..., \hat{q_m}\}$ and m ground-truth subjective quality scores $Q = \{q_1, q_2, ..., q_m\}$.

Specifically, the monotonicity-induced loss predicts the monotonicity of the video quality scores by introducing additional order constraints. The monotonicity-induced loss function is defined as follows:

$$L_{mon} = \frac{1}{m^2} \sum_{i=1}^{m} \sum_{j=1}^{m} max(0, |q_i - q_j| - f(q_i, q_j) \cdot (\hat{q_i} - \hat{q_j})) \quad (19)$$

where $f(q_i, q_j) = 1$ if $q_i \geq q_j$, otherwise $f(q_i, q_j) = -1$.

In contrast, the goal of the linearity-induced loss is to compute the linear relationship between the predicted quality score and ground-truth subjective quality score. The linearity-induced loss function can be denoted as:

$$L_{lin} = (1 - \frac{\sum_{i=1}^{m}(\hat{q_i} - \hat{a})(q_i - a)}{\sqrt{\sum_{i=1}^{m}(\hat{q_i} - \hat{a})^2 \sum_{i=1}^{m}(q_i - a)^2}})/2 \quad (20)$$

where $a = \frac{1}{m} \sum_{i=1}^{m} q_i$ and $\hat{a} = \frac{1}{m} \sum_{i=1}^{m} \hat{q_i}$.

Finally, the total loss function $L$ is obtained by combining the two loss functions $L_{mon}$ and $L_{lin}$ above:

$$L = \alpha L_{mon} + \beta L_{lin} \quad (21)$$

where $\alpha$ and $\beta$ represent the weights of monotonicity-induced loss and linearity-induced loss.

## 5 EXPERIMENTS

### 5.1 Experimental Setups

*5.1.1 Datasets.* We test the model on four datasets including LSVQ [70], KoNViD-1k (1200 videos) [16], LIVE-VQC (585 videos) [48], and YouTube-UGC (1067 videos) [56]. Specifically, we pre-train CLiF-VQA on $LSVQ_{train}$, a subset of LSVQ containing 28,056 videos. Intra-dataset testing is performed on two subsets of LSVQ, $LSVQ_{test}$ (7400 videos) and $LSVQ_{1080p}$ (3600 videos). We perform cross-dataset testing on KoNViD-1k and LIVE-VQC. Further, we fine-tune the model on KoNViD-1k, LIVE-VQC, and YouTube-UGC. It should be noted that YouTube-UGC contains 1500 videos, but only 1067 videos are available to us.

*5.1.2 Evaluation Criteria.* Spearman Rank Order Correlation Coefficient (SROCC) and Pearson Linear Correlation Coefficient (PLCC) are used as evaluation Metrics. Specifically, SROCC is used to measure the prediction monotonicity between predicted scores and true scores by ranking the values in both series and calculating the linear correlation between the two ranked series. In contrast, PLCC evaluates prediction accuracy by calculating the linear correlation between a series of predicted scores and true scores. And higher SROCC and PLCC scores indicate better performance.

*5.1.3 Implementation Details.* we employ PyTorch framework and an NVIDIA GeForce RTX 3090 card to train the model in all experimental implementations. In the semantic feature extraction module, we sample each frame of the video 9 times and then perform zero-shot feature extraction with CLIP, as shown in Fig. 6(b). In the spatio-temporal feature extraction module, we use Video Swin Transformer Tiny [36], pre-trained on the Kinetics-400 [24] dataset, as the backbone. During training, the initial learning rate of Swin Transformer backbone is set to 0.000075, and the initial learning of other parts is set to 0.00075. We set the batch size to 12 and use the AdamW optimizer with a weight decay rate of 0.05.

*5.1.4 Baseline Methods.* We compare the proposed method with the following methods:

- Classical Methods: BRISQUE [40], TLVQM [25], VIDEVAL [53], RAPIQUE [54].
- Classical + Deep Learning Methods: CNN+TLVQM [26], CNN+VIDEVAL [53].
- Deep Learning Methods: VSFA [29], PVQ [70], BVQA [28], GST-VQA [3], CoINVQ [57], FAST-VQA [58], FasterVQA [59], DOVER [63].

**Table 1: Experimental performance of the pre-trained CLiF-VQA model on the LSVQ dataset on four test sets (LSVQ$_{test}$, LSVQ$_{1080p}$, KoNViD-1k, LIVE-VQC). LSVQ$_{test}$ and LSVQ$_{1080p}$ are used for intra-dataset testing. While KoNViD-1k and LIVE-VQC are used for cross-dataset testing. Best in red and second in blue.**

| Testing Type | | Intra-dataset Test Datasets | | | | Cross-dataset Test Datasets | | | |
|---|---|---|---|---|---|---|---|---|---|
| Testing Datasets | | LSVQ$_{test}$ | | LSVQ$_{1080p}$ | | KoNViD-1k | | LIVE-VQC | |
| Methods | Source | SROCC | PLCC | SROCC | PLCC | SROCC | PLCC | SROCC | PLCC |
| BRISQUE [40] | TIP, 2012 | 0.569 | 0.576 | 0.497 | 0.531 | 0.646 | 0.647 | 0.524 | 0.536 |
| TLVQM [25] | TIP, 2019 | 0.772 | 0.774 | 0.589 | 0.616 | 0.732 | 0.724 | 0.670 | 0.691 |
| VIDEVAL [53] | TIP, 2021 | 0.794 | 0.783 | 0.545 | 0.554 | 0.751 | 0.741 | 0.630 | 0.640 |
| VSFA [29] | ACMMM, 2019 | 0.801 | 0.796 | 0.675 | 0.704 | 0.784 | 0.794 | 0.734 | 0.772 |
| PVQ$_{wo/patch}$ [70] | CVPR, 2021 | 0.814 | 0.816 | 0.686 | 0.708 | 0.781 | 0.781 | 0.747 | 0.776 |
| PVQ$_{w/patch}$ [70] | CVPR, 2021 | 0.827 | 0.828 | 0.711 | 0.739 | 0.791 | 0.795 | 0.770 | 0.807 |
| BVQA [28] | TCSVT, 2022 | 0.852 | 0.854 | 0.771 | 0.782 | 0.834 | 0.837 | 0.816 | 0.824 |
| FAST-VQA-M [58] | ECCV, 2022 | 0.852 | 0.854 | 0.739 | 0.773 | 0.841 | 0.832 | 0.788 | 0.810 |
| FAST-VQA [58] | ECCV, 2022 | 0.872 | 0.874 | 0.770 | 0.809 | 0.864 | 0.862 | 0.824 | 0.841 |
| FasterVQA [59] | TPAMI, 2023 | 0.873 | 0.874 | 0.772 | 0.811 | 0.863 | 0.863 | 0.813 | 0.837 |
| DOVER [63] | ICCV, 2023 | 0.881 | 0.879 | 0.782 | 0.827 | 0.871 | 0.872 | 0.812 | 0.841 |
| **CLiF-VQA** | **Ours** | 0.886 | 0.887 | 0.790 | 0.832 | 0.877 | 0.874 | 0.834 | 0.855 |
| *improvement to existing best* | | 0.57% | 0.91% | 1.02% | 0.61% | 0.69% | 0.23% | 1.21% | 1.66% |

**Table 2: The finetune results on LIVE-VQC, KoNViD and YouTube-UGC. Best in red and second in blue.**

| Finetune Datasets | | LIVE-VQC(585) | | KoNViD-1k(1200) | | YouTube-UGC(1067) | | Average | |
|---|---|---|---|---|---|---|---|---|---|
| Methods | Source | SROCC | PLCC | SROCC | PLCC | SROCC | PLCC | SROCC | PLCC |
| TLVQM [25] | TIP, 2019 | 0.799 | 0.803 | 0.773 | 0.768 | 0.669 | 0.659 | 0.732 | 0.726 |
| VIDEVAL [53] | TIP, 2021 | 0.752 | 0.751 | 0.783 | 0.780 | 0.779 | 0.773 | 0.772 | 0.772 |
| RAPIQUE [54] | OJSP, 2021 | 0.755 | 0.786 | 0.803 | 0.817 | 0.759 | 0.768 | 0.774 | 0.790 |
| CNN+TLVQM [26] | ACMMM, 2020 | 0.825 | 0.834 | 0.816 | 0.818 | 0.809 | 0.802 | 0.815 | 0.814 |
| CNN+VIDEVAL [53] | TIP, 2021 | 0.785 | 0.810 | 0.815 | 0.817 | 0.808 | 0.803 | 0.806 | 0.810 |
| VSFA [29] | ACMMM, 2019 | 0.773 | 0.795 | 0.773 | 0.775 | 0.724 | 0.743 | 0.752 | 0.765 |
| PVQ [70] | CVPR, 2021 | 0.827 | 0.837 | 0.791 | 0.786 | NA | NA | NA | NA |
| GST-VQA [3] | TCSVT, 2021 | NA | NA | 0.814 | 0.825 | NA | NA | NA | NA |
| CoINVQ [57] | TCSVT, 2021 | NA | NA | 0.767 | 0.764 | 0.816 | 0.802 | NA | NA |
| BVQA [28] | TCSVT, 2022 | 0.831 | 0.842 | 0.834 | 0.836 | 0.831 | 0.819 | 0.832 | 0.832 |
| FAST-VQA-M [58] | ECCV, 2022 | 0.803 | 0.828 | 0.873 | 0.872 | 0.768 | 0.765 | 0.815 | 0.822 |
| FAST-VQA [58] | ECCV, 2022 | 0.845 | 0.852 | 0.890 | 0.889 | 0.857 | 0.853 | 0.864 | 0.865 |
| FasterVQA [59] | TPAMI, 2023 | 0.843 | 0.858 | 0.895 | 0.898 | 0.863 | 0.859 | 0.867 | 0.872 |
| DOVER [63] | ICCV, 2023 | 0.812 | 0.852 | 0.897 | 0.899 | 0.877 | 0.873 | 0.862 | 0.875 |
| **CLiF-VQA** | **Ours** | 0.866 | 0.878 | 0.903 | 0.903 | 0.888 | 0.890 | 0.886 | 0.890 |
| *improvement to existing best* | | 2.49% | 2.33% | 0.67% | 0.45% | 1.25% | 1.95% | 2.19% | 1.71% |

**Table 3: FLOPs and running time(average of 10 runs) on GPU (RTX 3090) and CPU (i7-14700K) comparison of CLiF-VQA.**

| Methods | 540p | | | 720p | | | 1080p | | |
|---|---|---|---|---|---|---|---|---|---|
| | FLOPs(G) | Time(GPU/s) | Time(CPU/s) | FLOPs(G) | Time(GPU/s) | Time(CPU/s) | FLOPs(G) | Time(GPU/s) | Time(CPU/s) |
| VSFA [29] | 6440 | 1.506 | 38.65 | 11426 | 2.556 | 64.66 | 25712 | 5.291 | 150.2 |
| PVQ [70] | 9203 | 1.792 | 39.71 | 13842 | 2.968 | 68.50 | 36760 | 6.556 | 173.7 |
| BVQA [28] | 17705 | 3.145 | 101.5 | 31533 | 7.813 | 165.8 | 70714 | 14.34 | 510.6 |
| FAST-VQA [58] | 284 | 0.246 | 4.383 | 284 | 0.246 | 4.297 | 284 | 0.248 | 4.338 |
| DOVER [63] | 282 | 0.310 | 6.098 | 282 | 0.310 | 6.259 | 282 | 0.310 | 6.139 |
| **CLiF-VQA** | 1432 | 1.395 | 33.26 | 1432 | 1.397 | 33.31 | 1432 | 1.394 | 33.24 |

## 5.2 Pre-training Results on LSVQ

We pre-train CLiF-VQA on LSVQ and compare it with the existing advanced classical and deep VQA methods on four test datasets, as shown in Tab. 1. All experiments are conducted under 10 train-test splits. Compared with some classical methods, CLiF-VQA achieves a significant improvement in performance on all test datasets. In addition, CLiF-VQA achieves better results compared to FAST-VQA and FasterVQA, which focus only on low-level-aware features of the video. This suggests that the introduced human feelings features can well complement the spatial features, thus improving the prediction

accuracy. In addition, CLiF-VQA performs better on both intra-dataset testing and cross-dataset testing compared to the current state-of-the-art DOVER, with an average improvement of 1.02% and 0.85% on SROCC and PLCC.

## 5.3 Fine-tuning Results on Small Datasets

After pre-training on LSVQ, we fine-tune CLiF-VQA on three small datasets (LIVE-VQC, KoNViD-1k, YouTube-UGC), as shown in Tab. 2. As before, all experiments are conducted under 10 train-test splits. As can be seen, CLiF-VQA achieves unprecedented performance on

all three datasets. Relative to the current best performance, CLiF-VQA improved by an average of 2.19% and 1.71% on SROCC and PLCC, respectively. The results further illustrate the effectiveness of introducing human feelings in VQA.

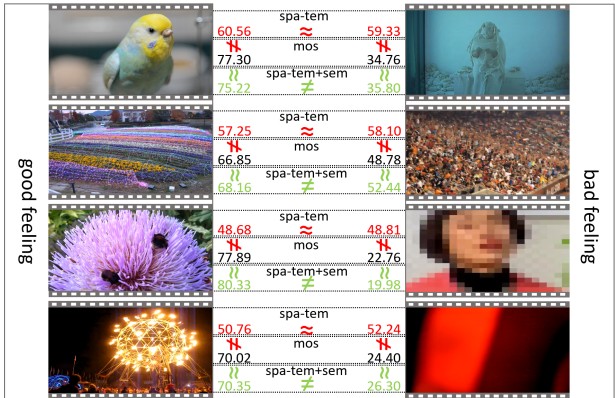

**Figure 7: The performance of the model on two sets of videos when using only spatio-temporal features as well as using both spatio-temporal and semantic features.**

## 5.4 Efficiency

To test the efficiency, we compare CLiF-VQA with several deep learning-based VQA methods, as shown in Tab. 3. Specifically, we compare the FLOPs and GPU/CPU runtimes for videos of different resolutions, where the length of the videos are 150 frames. **Since the semantic feature extraction process is performed offline, our model does not include the parameters of CLIP during training, and thus does not add too much computational effort.** In the efficiency test, for a fair comparison, we consider the increase in FLOPs and computation time due to the offline use of CLIP for extracting video semantic features. Compared with VSFA, PVQ, and BVQA, CLiF-VQA reduces FLOPs by up to 18x, 26x, and 49x, as well as reduces computation time by up to 4x, 5x, and 15x, respectively. In addition, CLiF-VQA achieves the best performance with acceptable FLOPs and computation time compared to the fastest VQA methods, FAST-VQA and DOVER.

**Table 4: Ablation study of three main components: Semantic Feature Extraction, Spatio-Temporal Feature Extraction and Regression Head. SROCC and PLCC are are average results on LIVE-VQC, KoNViD-1k and YouTube-UGC.**

| Semantic | Spatial | Regression | SROCC | PLCC |
|:---:|:---:|:---:|:---:|:---:|
| ✓ | | | 0.792 | 0.788 |
| | ✓ | | 0.864 | 0.865 |
| ✓ | | ✓ | 0.812 | 0.820 |
| | ✓ | ✓ | 0.868 | 0.869 |
| ✓ | ✓ | | 0.879 | 0.882 |
| ✓ | ✓ | ✓ | 0.886 | 0.890 |

## 5.5 Ablation Studies

*5.5.1 Ablation on the Compositions of CLiF-VQA.* We validate the effectiveness of the three modules that make up CLiF-VQA. As shown in Tab. 4, CLiF-VQA has acceptable performance when only semantic features related to human feelings are extracted, and the performance of CLiF-VQA is further improved when the regression head is introduced. When only low-level-aware features of

the video are extracted, CLiF-VQA performs better than when only semantic features are extracted. However, the performance did not improve significantly after further introducing the regression head. When features related to human feelings are introduced on top of the spatial features, the performance of the model improves significantly, and it is further improved by introducing the regression head. In addition, we further compare the performance of the model on videos that elicit different feelings in humans when using only spatial features and when using both spatial and semantic features, as shown in Fig. 7. We choose two sets of videos that have very different MOS, but have similar quality scores when predicted using only spatial features. After we further introduce semantic features related to human feelings, the predicted quality scores are closer to the real MOS. These experimental results illustrate the validity of human feelings we introduced in VQA.

**Table 5: Ablation study on descriptions. 'Obj' and 'Sub' denote objective and subjective descriptions, respectively.**

| Datasets | LIVE-VQC | KoNViD-1k | YouTube-UGC |
|---|---|---|---|
| Descriptions | SROCC/PLCC | SROCC/PLCC | SROCC/PLCC |
| None | 0.845/0.852 | 0.890/0.889 | 0.857/0.853 |
| Only-Obj | 0.857/0.868 | 0.898/0.895 | 0.880/0.876 |
| Only-Sub | 0.849/0.856 | 0.893/0.891 | 0.863/0.860 |
| Obj+Sub | 0.866/0.878 | 0.903/0.903 | 0.888/0.890 |

*5.5.2 Ablation on Descriptions.* In Tab. 5, we verify the effect of different types of descriptions on the performance of CLiF-VQA. The results demonstrate that objective descriptions have a greater impact on the performance improvement of CliF-VQA compared to subjective descriptions. And the optimal results can be obtained by using both objective and subjective descriptions.

## 6 CONCLUSION

In this paper, we first analyze that human feelings have a significant impact on video quality assessment (VQA). Further, We validate for the first time that CLIP is highly consistent with human feelings in video quality perception. Extensive experiments demonstrate that CLIP not only has good consistency with human feelings, but also can achieve satisfactory results in VQA by using only the features related to human feelings extracted by CLIP. Motivated by these findings, we propose CLiF-VQA, a method that extracts features related to human feelings and low-level-aware features of the video. Then, the quality score of the video is obtained by aggregating the two features. Experimental results demonstrate that the proposed CLiF-VQA outperforms existing methods on multiple VQA datasets.

## 7 LIMITATION

Our model is not end-to-end, and since considering CLIP's in training would lead to slow computation, we perform CLIP feature extraction separately. There is room for optimizing the prompts we design, such as increasing the number of prompts as well as and selecting better prompts.

## ACKNOWLEDGMENTS

This work was supported in part by the National Key R&D Program of China (2021YFF0900500), and the National Natural Science Foundation of China (NSFC) under grants 62441202 and U22B2035.

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
