# OpenReview forum: "CLiF-VQA: Enhancing Video Quality Assessment by Incorporating High-Level Semantic Information related to Human Feelings"
_acmmm.org/ACMMM/2024/Conference — MM2024 Poster_

### Official Review · Reviewer_LWpj · 2024-04-28

**Rating:** 2
**Confidence:** 4

**Summary:**

This paper explores the application of CLIP in the field of video quality assessment.
The paper utilizes CLIP to extract high-level semantic information and employs prompts related to human feeling, which enhances the performance of existing VQA.

**Strengths:**

The article explores the ability of CLIP to effectively extract human feeling features and does some subjective testing.
Meanwhile, I think there are some shortcomings in the novelty, theory, and experimentation of the paper, and these shortcomings lead me to not recognize the innovation of the paper. Please refer to **Limitations**.

**Limitations:**

Major issue:
In fact, the paper lacks theoretical explanations as well as sound logical descriptions. I will show my point by listing these aspects as follows:
(1) The paper does not explain the correlation between subjective quality scores and human feeling, including not even defining human feeling. This means that even though CLIP extracts human feeling features, it does not effectively represent video quality.
(2) When there is some correlation between subjective quality and subjective perception, the novelty of the article will be severely lacking. Because, the pipeline of the paper is almost the same as [1].
(3) In Figure 5, the results of “Only subjective” is completely inconsistent with subjective quality, does this mean that human feeling is not correlated with subjective quality.
(4) In Figure 6, the high-level semantics will be lost after the video frames are divided into blocks.
(5) The ablation experiments did not even use ViT (pre-trained in ImageNet) + spatial-temporal as a comparison to illustrate that the gain comes from CLIP rather than semantic features.
Based on these shortcomings, I think the novelty of the paper is limited.

Other issue:
**Introduction:**
(1) Lines 112-115 "It’s pretty obvious that human ... different subjective quality scores.", This sentence is difficult to understand because different content has different subjective qualities but they have the same objective quality, couldn't this be due to poor performance of the objective model.
(2) Lack of definition of "human feeling", including Figure 1.

**Related Work:**
(1) The paper uses CLIP to extract high-level semantic features, yet there is no information describing CLIP in related work (or Sections 3 or 4)
(2) What is the difference between the proposed method and MaxVQA [1]. As far as I know, MaxVQA also uses a combination of CLIP + Fast-VQA.

**Section 3:**
The writing is confusing.
First, the first description of Figure 3 precedes Figure 2.
Second, what is the difference between the "+" in the third and fourth columns of figure 3, and what do the numbers mean?
Where does the data source for Figure 4 come from, and where does the human feeling labeling come from.
In Figure 5, the KROCC is not defined and it is clear from Figure 5 that the objective description is more facilitative. Also, CLIP uses ViT as a backbone network, while the comparison method is based on CNN.

**Proposed method:**
The pipeline of the proposed method is very similar to MaxVQA, except for the addition of human feeling prompts. However, the paper does not have a reliable theoretical or experimental proof that human sensory cue words have a role in video quality.

**Experiments:**
(1) The comparison with MaxVQA is missing, due to the fact that both the proposed method and MaxVQA are based on multi-modal VQA models, while the comparison method is based on unimodal.
(2) Lines 687-689, "Limited by our video memory (24G), we set the batch size to 12 when reproducing FAST-VQA [46] as well as DOVER [51], instead of the 16 used in the original paper", changing the hyperparameters of the comparison method due to memory limitations is a very imprecise experimental setup. To my knowledge, setting the batch size of DOVER [2] to 12 still cannot run on RTX 3090.

[1] Haoning Wu, Erli Zhang, Liang Liao, Chaofeng Chen, Jingwen Hou, Annan Wang, Wenxiu Sun, Qiong Yan, and Weisi Lin. 2023. Towards Explainable In-the-Wild Video Quality Assessment: A Database and a Language-Prompted Approach. In ACM MM. 1045–1054.
[2] Haoning Wu, Erli Zhang, Liang Liao, Chaofeng Chen, Jingwen Hou, Annan Wang, Wenxiu Sun, Qiong Yan, and Weisi Lin. 2023. Exploring video quality assessment on user generated contents from aesthetic and technical perspectives. In ICCV. 20144–20154.

**Suitability:**

3

---

### Official Review · Reviewer_Hjoi · 2024-05-25

**Rating:** 3
**Confidence:** 3

**Summary:**

The paper proposes a novel method called CLiF-VQA to enhance video quality assessment by incorporating high-level semantic information related to human feelings. The key contributions are:
Validating for the first time that Contrastive Language-Image Pre-training (CLIP) is highly consistent with human feelings in video perception.
Proposing CLiF-VQA, which for the first time incorporates features related to human feelings in VQA, achieving state-of-the-art performance on multiple VQA datasets.
Designing effective prompts, including objective and subjective descriptions closely related to human feelings, as linguistic prompts for CLIP.
Developing a zero-shot advanced semantic feature extractor (SFE) based on CLIP to extract semantic features from videos.

**Strengths:**

(1) The methodology of incorporating human feelings features into VQA is innovative and practical, leveraging the capabilities of MLLMs like CLIP.
(2) Extensive experiments demonstrate the effectiveness of CLiF-VQA, achieving significant improvements over existing methods on multiple datasets.
(3) The use of CLIP and the designed prompts provide insights into how linguistic expressions can be utilized to capture feelings elicited by video content.
(4) The ablation studies validate the contributions of different components of CLiF-VQA and demonstrate the importance of incorporating human feelings features.

**Limitations:**

(1) The implementation details, especially regarding the hyperparameters and training procedures, are not provided in sufficient detail to fully reproduce the results.
(2) More thorough comparisons and discussions with related works, especially recent ones, could strengthen the evaluation of CLiF-VQA.
(3) Further ablation studies exploring the impact of different design choices (e.g., prompt selection) could provide additional insights.
(4) The theoretical analysis of why incorporating human feelings improves VQA could be expanded to deepen the understanding of the proposed approach.
(5) As far as I konw, the model metrics is lower than other experiments (e.g the dover's metrics on KoNViD-1k)

**Suitability:**

2

---

### Official Review · Reviewer_KLUR · 2024-05-26

**Rating:** 5
**Confidence:** 4

**Summary:**

This paper proposes a new VQA approach, CLiF-VQA, by integrating FAST-VQA and CLIP feeling modules for better Video Quality Assessment performance. It claims that human feelings on videos will also affect the final quality scores of videos, and hence proposes to include "feeling-prompts" into the VQA model. The CLiF-VQA suggests better accuracy than previous models.

**Strengths:**

1. It is clearly motivated.
2. The writing is good, well understandable; the equations are well-defined.
3. The evaluation is sufficient, proving effectiveness of CLiF-VQA.

**Limitations:**

1. CLIP is not an MLLM. Please try to revise this misleading name.
2. Please try to discuss the limitations of this work.

**Suitability:**

3

---

### Official Review · Reviewer_PPPE · 2024-05-28

**Rating:** 4
**Confidence:** 3

**Summary:**

In this paper, the authors propose a video quality assessment method, called CLiF-VQA, which augments traditional spatio-temporal feature assessment by incorporating a multimodal large language model, CLIP, to extract high-level semantic information related to human emotions. Through extensive experimental validation on multiple datasets, CLiF-VQA shows highly consistent results with humans in video quality perception, with a promising performance improvement over existing methods.However, this paper lacks a detailed discussion of the potential limitations in practical applications and the specific advantages and disadvantages of the method.However, this paper lacks a detailed discussion of the potential limitations in practical applications and the specific advantages and disadvantages of the method. However, this paper lacks a detailed discussion of the potential limitations and the specific advantages and disadvantages of the method.

**Strengths:**

（1）	The method pioneers the exploration of the consistency between CLIP and human feelings in video perception, offering a new dimension to VQA by considering both objective and subjective aspects.
（2）	The approach demonstrates improved performance over existing methods by combining traditional spatio-temporal features with high-level semantic features.
（3）	Figures and tables are relevant and well-designed, providing clear visual support to the text. For example, Figure 1 effectively illustrates the impact of human feelings on VQA.
（4）	The experimental setup is robust, involving extensive testing on multiple datasets, including LSVQ, KoNViD-1k, LIVE-VQC, and YouTube-UGC.

**Limitations:**

（1）	While the integration of human feelings is innovative, the practical implications and potential limitations in various real-world applications are not thoroughly discussed.
（2）	Some figure captions could be more descriptive to provide better context. For example, the caption for Figure 3 could include more details about the significance of the results shown.
（3）	The discussion of experimental results could be expanded to include more insights into the limitations and potential areas for improvement of the proposed method.

**Suitability:**

2

---

### Meta-Review · Area_Chair_dCzC · 2024-07-08

**Recommendation:** Accept (Poster)
**Confidence:** 4

**Metareview:**

Borderline rating from the reviewers, which are all quite confident, but looking at their background, most are at the start of their scientific careers, so not sure that the confidence level should be so high. Examined the paper myself and I agree with the most senior reviewer's recommendation.